

# Sensitivity of cross-sectional compliance to manufacturing tolerances for wind turbine blades

Vincent K. Maes[1], Terence Macquart[1], Paul M. Weaver[1,2], and Alberto Pirrera[1]

[1]Bristol Composite Institute, University of Bristol, UK
[2]Bernal Institute, University of Limerick, Ireland

**Correspondence:** Vincent K. Maes (vincent.maes@bristol.ac.uk)

**Abstract.** Wind-turbine blades are complex structures and, despite advancements in analysis techniques, differences persist between predictions of their elastic response and experimental results. This undermines confidence in the ability to reliably design and certify novel blade designs that include self-regulating features like bend-twist coupling To address these discrepancies, this study investigates the influence of manufacturing tolerances on the compliance properties of blade cross-sections,
focusing specifically on a previously disregarded feature: the trailing edge bond-line. To conduct this investigation, the validated cross-sectional modelling tools BECAS and VABS are used to demonstrate that even small geometric variations can have significant influence on cross-sectional stiffness properties. The results are further examined and substantiated through the utilisation of 3D finite element models, adopting both shell and solid elements. We conclude that an accurate geometric representation of the cross-section is necessary to adequately capture the shear flow within it and assure accurate predictions
on cross-sectional stiffness properties.

## 1 Introduction

Over the past few decades, with an ever increasing focus on climate change, the wind energy industry has seen growth in both cumulative installed power and the size of individual wind turbines McKenna et al. (2016). The increasing length of blades exacerbates key load cases, such as gust and fatigue loads, with cascading implications. Higher loads indeed result in increased
root bending moments, which are transmitted to the nacelle, necessitating heavier and more costly designs for the generator and tower components of the wind turbine. To address this issue, the Literature explores load alleviation strategies, both in passive and active form, as summarised by McKenna *et al.* McKenna et al. (2016).

Specifically, Bend-Twist Coupling (BTC) has been explored by several authors—among other passive adaptive solutions Ponta et al. (2014)—as a means of reducing the impact of peak (*i.e.* gust) and cyclical (*i.e.* fatigue) loads, thereby mitigating loads on
other turbine components and structures Bottasso et al. (2013); Vesel and McNamara (2014); Gözcü and Kayran (2014); Şener et al. (2017); Bagherpour et al. (2018); Manolas et al. (2018). As the name suggests, this aeroelastic tailoring principle operates through the elastic coupling of torsional and flap-wise bending motions of the blade (*i.e.*, a pure torsional load produces bending deflections, and a pure bending load produces twist), and achieves the goal of load alleviation most commonly by





coupling flap-wise deflection with twist towards feather. This twist causes a reduction of the aerodynamic loads, establishing
in a self-regulating system.

Previous optimisation studies have investigated the benefits of BTC Bottasso et al. (2013); Capuzzi et al. (2014, 2015);
Scott et al. (2016, 2017); Bagherpour et al. (2018); Chen et al. (2019); Wiens et al. (2020); Serafeim et al. (2022). Three
distinct approaches to achieve BTC have been examined: geometric coupling, material coupling, and combined coupling. In
this work the focus is on the application of material coupling, as enabled by composite materials, which, in addition to their
excellent specific (*i.e.* per unit mass) properties, offer significant stiffness tailoring capabilities due to their anisotropy. With
the stiffness in the fibre direction being generally many times greater than in the other directions, composite plies can be used
to effectively induce a wide range of structural couplings. However, integration of non-standard angle plies within a structure
raises many questions concerning manufacturability, as well as the strength properties at the laminate and blade scale. This
is because material allowables are usually acquired through costly test campaigns, which is why there is not much data for
non-conventional angles.

While the benefits of material BTC have been well documented, its commercial uptake remains limited, prompting the
question of why it has not seen widespread adoption. Currently, only two publicly documented examples of structures using
BTC exist: the Grumman X-29 Greenhalgh et al. (1993); Pamadi (2015) and the Westland BERP IV helicopter blade Harrison
et al. (2008); Moffatt and Griffiths (2009). Both of these aerospace structures demonstrate the capabilities of BTC; however,
publicly available documentation on their operating performance is scarce. In addition to the uncertainties on the strength
characteristics as described above, the limited industrial uptake of BTC may also be attributed to uncertainties surrounding the
quantification of the stiffness of structures comprising coupled laminates. Focusing specifically on the wind energy field, while
some studies have carried out experimental tests on the elastic performance of BTC blades, the results have been inconsistent.
Significant discrepancies in predicted cross-sectional stiffness properties from different modelling and analysis schemes raise
questions regarding the validity of the methods available Chen et al. (2010); Saravia et al. (2017); Lekou et al. (2015). It is
important to note that this issue extends beyond BTC blades, and may be attributed to inconsistencies in the pre-processing
steps preceding numerical analysis Lekou et al. (2015). This work aims to address the previously observed discrepancies in
cross-sectional properties between different models by investigating the sensitivity of numerical models for stiffness prediction
to generally overlooked, detailed cross-sectional features.

A study by Lekou *et al.* Lekou et al. (2015) revealed significant variations in numerically predicted stiffness performance
metrics when the same blade description was provided to six different design and research teams. The authors of the study point
out that many of the methods employed by the teams had been cross-validated for simpler geometries. This suggests that, as
designs become more intricate, individual interpretation and handling of modelling inputs can lead to discrepancies. However,
these discrepancies only become critical or even discernible, if the structure's performance is sensitive to them. A BTC blade
with high bending stiffness, for instance, will by its own nature not deflect much under loads and therefore experience minimal
changes in torsional deflections, and hence performance, even if it is a few percent stiffer or more compliant than originally
designed. Conversely, for a BTC blade with greater bending compliance, a slight variation in bending deflections can result
in significant differences in torsional deflections and aeroelastic performance. For similar reasons, inaccuracies in torsional or





bend-twist coupling stiffnesses have historically been uninfluential and have only emerged as a problem with the increase in
rotors diameter and blade slenderness.

From previous work it is known that BTC is driven by the development of shear flow, and, as studied experimentally
by Lemanski (2004), the resultant twisting behaviour can be heavily influenced by any changes in said stress distribution.
Lemanski and Weaver Lemanski and Weaver (2005) showed the importance of the shear flow continuity between flanges
and webs, and the influence of the restraint put on the flange's deformation by the presence of the webs. Their work clearly
demonstrated the improvement in prediction of the coupling stiffness terms obtained by accounting for these two effects. Their
work was extended by Canale *et al.* Canale et al. (2018) to calculating the bending and torsional stiffnesses of box sections using
the same considerations. Again, a significant improvement in agreement with finite element method results was demonstrated
as a result of including the shear flow continuity and constraint effects of the webs on the deformation of the flanges. In light
of this, shear flow behaviour in the scenarios studied will be considered in this work.

The main goal, however, is to further evaluate the influence of different geometrical variations on the stiffness properties of
blade designs incorporating BTC. To this end, a set of representative wind turbine cross-sections are run through a sensitivity
analysis, employing BECAS and VABS. The results in this paper include the variability of four compliance terms of key
interest in this work for their relation to BTC: the bending compliance; the bend-twist coupling compliance; the torsional
compliance and the shear compliance. These are the four compliance terms that drive the bend-twist coupling behaviour of a
blade. However, the analysis performed does provide the full stiffness/compliance matrices and hence readers seeking data on
the remaining terms may refer to Maes (2021).

## 2 Model definition

To investigate the influence of manufacturing tolerances on stiffness properties, three cross-sections were generated. These
cross-sections were designed to be representative of industrial blades in a general sense, without replicating any particular
design, using a combination of unidirectional glass fibre epoxy (UD), bi-axial glass fibre epoxy (BIAX), structural foam and
epoxy adhesive materials. Exact manufacturing tolerances are of course process dependent, and the designs and tolerances in
this study are based on the most commonly adopted blade architecture, where suction-side and pressure-side sandwich skins
form the aerodynamic shape of the structure and are bonded together encasing one or more internal webs. Other manufacturing
processes will lead to different tolerances and would require separate analyses.

### 2.1 Geometry

The cross-sections under investigation are shown in Figure 1, representing three stations along the length of a typical blade,
with different thickness-to-chord ratios and relative material distributions.

They are segmented into regions to which material properties are assigned, as illustrated in Figure 2. The segmentation
allows for parameterisation of the cross-section by defining the chordwise locations of the boundaries of each region. The



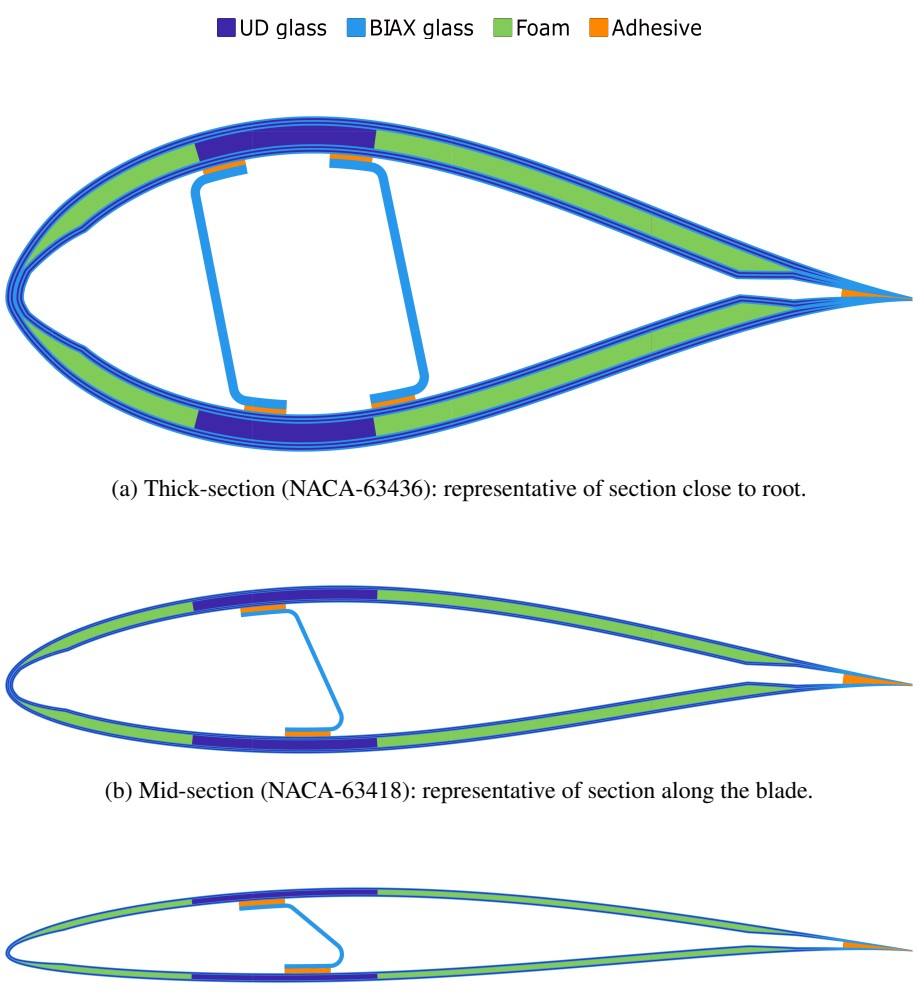

(a) Thick-section (NACA-63436): representative of section close to root.

(b) Mid-section (NACA-63418): representative of section along the blade.

(c) Thin-section (NACA-63410): representative of section close to tip.

**Figure 1.** Cross-sections of the base sections, showing material distribution, as generated by BECAS. In the legend, UD stands for unidirectional material, BIAX for bi-axial material.





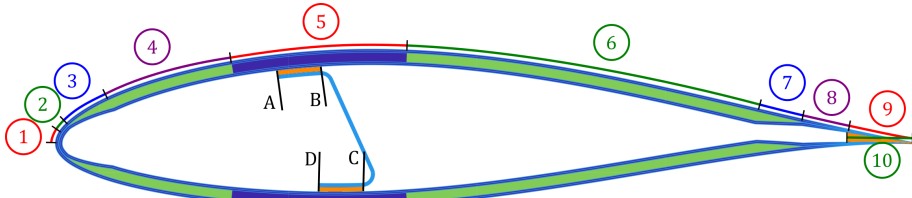

**Figure 2.** Mid cross-section indicating regions of blade, including contact points for webs, which are identically located along the chordwise direction for both top and bottom surface.

**Table 1.** Bounds for regions in Figure 2, which are identical for all cross-sections. All dimensions relative to chord length, which is zeroed at leading edge (LE). The trailing edge (TE) is trimmed at 0.975 of chord length.

| Region | Bounds | | |
|---|---|---|---|
| (1) Tip Transition | 0.000 | - | 0.005 |
| (2) Tip Coupled | 0.005 | - | 0.010 |
| (3) Core Ramp Up | 0.010 | - | 0.060 |
| (4) LE Core | 0.060 | - | 0.200 |
| (5) Spar Cap | 0.200 | - | 0.400 |
| (6) TE Core | 0.400 | - | 0.800 |
| (7) Core Ramp Down | 0.800 | - | 0.850 |
| (8) UD ply ramp-down | 0.850 | - | 0.900 |
| (9) BIAX ply ramp-down | 0.900 | - | 0.975 |
| (10) TE Bond depth | 0.900 | - | 0.975 |

baseline values for these boundary locations are provided in Table 1. Additionally, the corner radii (0.01 in all cases) and the locations of the web's contact points, as indicated in Table 2, are taken into account to complete the definition of each geometry.

**Table 2.** Chorwise location of contact points for webs in base configurations. Order is always (A) top surface start of web flange, (B) top surface contact point of web, (C) bottom surface contact point of web, (D) bottom surface end of web flange.

| Web | Contact Point | | | |
|---|---|---|---|---|
| | A | B | C | D |
| Thick Section: LE | 0.25 | 0.20 | 0.25 | 0.30 |
| Thick Section: TE | 0.35 | 0.40 | 0.45 | 0.40 |
| Mid Section | 0.25 | 0.30 | 0.35 | 0.30 |
| Thin Section | 0.25 | 0.30 | 0.35 | 0.30 |




The geometries studied here are not actually existing designs but are inspired by some older blade designs. While it is expected similar trends are likely to be observed in other designs, the exact values will be case specific. Two significant differences that the studied designs may have to other popular designs include (a) the number of blades, where modern large

blades will often have 2-3 webs running most, if not all, of the length of the blade, and (b) the manner in which webs are integrated into the spar caps. The studied designs represent a case where webs are separately manufactured and then bonded in, while some manufacturers may use an integral approach where the webs are directly co-cured into/onto the outer shell. Such variations experience both different levels of sensitivity and manufacturing tolerances to consider.

For the airfoil shapes, publicly available NACA profiles were used as indicated in Figure 1 for each section. All dimensional

inputs, including the cured ply thickness, are described relative to the chord length, which for the purpose of the results presented is $1\,\mathrm{m}$. The analyses used do not account for any scale dependent non-linear effects. This means that, as long as dimensions are kept proportional, the properties simply scale as a function of the chord length. Hence, the changes/sensitivities documented herein are not affected by the choice of chord length, as long as the other dimensions are scaled proportionally. For this reason, all dimensions in the $x$-axis of the plots in this paper have an asterisk added to them to remind the reader that

these changes are actually relative (*i.e.* can be read as percentage changes).

While in this study a clear position of the trailing edge bondline is given in the descriptions of the designs, this is rather unique. As many studies use shell elements, the trailing edge bondline is often not modelled at all. For example, the study by Branner *et al.* Branner et al. (2007), which looked specifically at the efficacy of shell element models *vs* shell-solid hybrid models, does not include a TE bondline. In general, top and bottom surfaces are either simply put into contact Chen et al.

(2010), connected at a single point Branner et al. (2007), or cut off in a flatback style configuration Lekou et al. (2015). The extensive study evaluating different structural analysis tools across multiple research groups by Lekou *et al.* Lekou et al. (2015) based its cross-sectional definitions on aerodynamic profiles and keypoints to indicate material distribution similar to the one used here, but left both the full geometry of the webs and the TE bondlines ambiguous. These observations motivate the choice in variations investigated in this work.

## 2.2  Material Properties and lay-ups

For material properties, generic values were utilised. Listed in Table 3, these values do not belong to any specific material but are roughly representative of the properties of materials used in industry.

BTC is designed into the mid-section and thin-section models—Figure 1b and Figure 1c respectively—by introducing UD plies within the skins (zones 4, 5, and 6 in Figure 2) that are at an angle to the beam axis (the out of the page direction). The

full lay-ups for each of the regions in the three cross-sections are provided in Table 4, Table 5, and Table 6. The web bondline thickness is $0.005$ for all locations and cross-sections. It should be noted that in regions 3 and 7, the foam core thickness is being ramped from (or to) zero, and so has a linearly varying thickness in those regions. In region 8 and 9, the UD and BIAX plies are similarly tapered in their thickness. For the UD plies in Region 8, this is done to ensure a smooth reduction in thickness, while for the BIAX plies in region 9, it is done to ensure no material overlap at the TE of the blade. The UD plies

are tapered to zero, while the BIAX plies are tapered to $1/8^{\text{th}}$ of their full ply thickness.

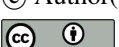



**Table 3.** Elastic properties for materials used for the numerical study, including Uni-Directional (UD) material, Bi-Axial (BIAX) material ($\pm 45$), Foam care material, and adhesive material used for bondlines.

| | | Material | | | |
|---:|:---:|:---:|:---:|:---:|:---:|
| Property | Units | UD glass | BIAX glass | Foam | Adhesive |
| $E_{11}$ | [GPa] | 35.0 | 10.0 | 0.075 | 3.50 |
| $E_{22} = E_{33}$ | [GPa] | 10.0 | 10.0 | 0.075 | 3.50 |
| $E_{12} = G_{13}$ | [GPa] | 4.0 | 10.0 | 0.025 | 1.30 |
| $G_{23}$ | [GPa] | 1.0 | 10.0 | 0.025 | 1.30 |
| $\nu_{12} = \nu_{13} = \nu_{23}$ | [-] | 0.3 | 0.3 | 0.45 | 0.35 |
| $t_{\mathrm{ply}}$ | [mm] | 0.25 | 0.25 | N.A. | N.A. |

**Table 4.** Layups for thick-section, for materials a shorthand is used; $U$ = UD at $0°$, $U^\alpha$ = UD at $20°$, $B$ = BIAX, with subscript indicating repeated plies and $F_X$ = FOAM of thickness to chord ratio $X$.

| Region | Layup |
|:---:|:---|
| (1) | $[B_8/U_8/B_8/U_8/B_{16}/U_8/B_8/U_8/B_8]$ |
| (2) | $[B_8/U_8^\alpha/B_8/U_8^\alpha/B_{16}/U_8^\alpha/B_8/U_8^\alpha/B_8]$ |
| (3) | $[B_8/U_8^\alpha/B_8/U_8^\alpha/B_8/F_{0.02}/B_8/U_8^\alpha/B_8/U_8^\alpha/B_8]$ |
| (4) | $[B_8/U_8^\alpha/B_8/U_8^\alpha/B_8/F_{0.02}/B_8/U_8^\alpha/B_8/U_8^\alpha/B_8]$ |
| (5) | $[B_8/U_8^\alpha/B_8/U_8^\alpha/B_8/U_{80}/B_8/U_8^\alpha/B_8/U_8^\alpha/B_8]$ |
| (6) | $[B_8/U_8^\alpha/B_8/U_8^\alpha/B_8/F_{0.02}/B_8/U_8^\alpha/B_8/U_8^\alpha/B_8]$ |
| (7) | $[B_8/U_8^\alpha/B_8/U_8^\alpha/B_8/F_{0.02}/B_8/U_8^\alpha/B_8/U_8^\alpha/B_8]$ |
| (8) | $[B_8/U_8^\alpha/B_8/U_8^\alpha/B_{16}/U_8^\alpha/B_8/U_8^\alpha/B_8]$ |
| (9) | $[B_{48}]$ |
| Webs | $[B_{40}]$ |




**Table 5.** Layups for mid-section, for materials a shorthand is used; $U$ = UD at $0°$, $U^\alpha$ = UD at $20°$, $B$ = BIAX with subscript indicating repeated plies and $F_X$ = FOAM of thickness to chord ratio $X$.

| Region | Layup |
|---|---|
| (1) | $[B_3/U_3/B_3/U_3/B_6/U_3/B_3/U_3/B_3]$ |
| (2) | $[B_3/U_3^\alpha/B_3/U_3^\alpha/B_6/U_3^\alpha/B_3/U_3^\alpha/B_3]$ |
| (3) | $[B_3/U_3^\alpha/B_3/U_3^\alpha/B_3/F_{0.01}/B_3/U_3^\alpha/B_3/U_3^\alpha/B_3]$ |
| (4) | $[B_3/U_3^\alpha/B_3/U_3^\alpha/B_3/F_{0.01}/B_3/U_3^\alpha/B_3/U_3^\alpha/B_3]$ |
| (5) | $[B_3/U_3^\alpha/B_3/U_3^\alpha/B_3/U_{40}/B_3/U_3^\alpha/B_3/U_3^\alpha/B_3]$ |
| (6) | $[B_3/U_3^\alpha/B_3/U_3^\alpha/B_3/F_{0.01}/B_3/U_3^\alpha/B_3/U_3^\alpha/B_3]$ |
| (7) | $[B_3/U_3^\alpha/B_3/U_3^\alpha/B_3/F_{0.01}/B_3/U_3^\alpha/B_3/U_3^\alpha/B_3]$ |
| (8) | $[B_3/U_3^\alpha/B_3/U_3^\alpha/B_6/U_3^\alpha/B_3/U_3^\alpha/B_3]$ |
| (9) | $[B_{18}]$ |
| Webs | $[B_{18}]$ |

**Table 6.** Layups for thin-section, for materials a shorthand is used; $U$ = UD at $0°$, $U^\alpha$ = UD at $20°$, $B$ = BIAX with subscript indicating repeated plies and $F_X$ = FOAM of thickness to chord ratio $X$.

| Region | Layup |
|---|---|
| (1) | $[B_3/U_3/B_6/U_3/B_3]$ |
| (2) | $[B_3/U_3^\alpha/B_6/U_3^\alpha/B_3]$ |
| (3) | $[B_3/U_3^\alpha/B_3/F_{0.005}/B_3/U_3^\alpha/B_3]$ |
| (4) | $[B_3/U_3^\alpha/B_3/F_{0.005}/B_3/U_3^\alpha/B_3]$ |
| (5) | $[B_3/U_3^\alpha/B_3/U_{20}/B_3/U_3^\alpha/B_3]$ |
| (6) | $[B_3/U_3^\alpha/B_3/F_{0.005}/B_3/U_3^\alpha/B_3]$ |
| (7) | $[B_3/U_3^\alpha/B_3/F_{0.005}/B_3/U_3^\alpha/B_3]$ |
| (8) | $[B_3/U_3^\alpha/B_6/U_3^\alpha/B_3]$ |
| (9) | $[B_{12}]$ |
| Webs | $[B_{18}]$ |





# 3 Analysis procedure and benchmarking

To analyse the proposed cross-sections and evaluate the sensitivities of cross-sectional parameters to variations in geometry, the cross-sectional modellers BECAS Blasques and Bitsche (2014) and VABS Hodges (2006); Yu et al. (2012) are used, as they are both well established, widely adopted tools. A parameterised script written in MATLAB is used to generate the input files

for both tools, submit the analysis, and extract the cross-sectional stiffness properties. The cross-sectional stiffness matrices are inverted to obtain the cross-sectional compliance properties as it is deemed more intuitive to assess changes in compliance as opposed to changes in stiffness. Especially due to the presence of coupling, it is easier to think of how pure bending load produces a certain amount of bending and twisting deflection (the compliance view) instead of how pure bending deflection would induce a mixture of bending and twisting reaction loads (the stiffness view). For the results presented, the inversions tells

us that if stiffness terms are used, opposite trends would be observed (*i.e.* if compliance correlates positively with a variation, stiffness would correlate negatively.)

To benchmark the cross-sectional analysers, BECAS and VABS, the mid-section reference geometry, *i.e.* the baseline configuration used as a reference in the sensitivity studies, was analysed using four different additional modelling approaches. These approaches include 3D FEM models in ABAQUS utilising Linear Shell Elements, Quadratic Shell Elements, Linear

Solid Elements, and Quadratic Solid Elements.

A single script was used to generate all the required geometries for the 2D cross-section, 3D shell, and 3D solid models, including the segmentation for assigning material properties. The 2D cross-section geometry was then used directly to generate BECAS and VABS models, while the 3D shell and 3D solid geometries were imported into ABAQUS to complete the model description. This approach ensured consistency in the geometries among the different solvers, minimising discrepancies that

could arise from separate geometry generation.

To extract cross-sectional properties from the 3D ABAQUS models, the 3D solution needs to be mapped onto a 1D beam model, based on either Euler-Bernoulli beam theory or Timoshenko beam theory equations. The equations used in this work, provided next, are similar to those previously derived by other authors in the literature Hill and Weaver (2004); Malcolm and Laird (2007), and use Timoshenko beam theory due the lower shear stiffness of composite materials making capturing the

shear effects explicitly relevant to the results. In order to back calculate the beam model properties, one starts with the basic $6 \times 6$ Timoshenko compliance matrix for a beam with six degrees of freedom—including two shear strains ($\gamma_{13}$ and $\gamma_{23}$), one extensional strain ($\epsilon_{33}$), two bending curvatures ($\kappa_1$ and $\kappa_2$), and one twist curvature ($\theta'_3$)—and the complementary sectional loads, including: two shear loads ($f_1$ and $f_2$), one extensional load ($f_3$), two bending moments ($m_1$ and $m_2$), and one torsional moment ($m_3$).



The cross-sectional deformation gradients and cross-sectional loads are related through the compliance matrix, $[S]$, as

$$
\begin{bmatrix} \gamma_{13} \\ \gamma_{23} \\ \epsilon_{33} \\ \kappa_1 \\ \kappa_2 \\ \theta_3' \end{bmatrix} = \begin{bmatrix} S_{11} & S_{12} & S_{13} & S_{14} & S_{15} & S_{16} \\ S_{12} & S_{22} & S_{23} & S_{24} & S_{25} & S_{26} \\ S_{13} & S_{23} & S_{33} & S_{34} & S_{35} & S_{36} \\ S_{14} & S_{24} & S_{34} & S_{44} & S_{45} & S_{46} \\ S_{15} & S_{25} & S_{35} & S_{45} & S_{55} & S_{56} \\ S_{16} & S_{26} & S_{36} & S_{46} & S_{56} & S_{66} \end{bmatrix} \begin{bmatrix} f_1 \\ f_2 \\ f_3 \\ m_1 \\ m_2 \\ m_3 \end{bmatrix}.
\tag{1}
$$

For a beam of length L, deforming linearly under tip loads, the linearised equations for deflections and rotations as a function of the position, $z$, along the beam as integrals of the cross-sectional deformation gradients above are

$$
\delta_1(z) = \int_0^z \gamma_{13} dz + \int_0^z \int_0^z \kappa_2 dz dz,
\tag{2a}
$$

$$
\delta_2(z) = \int_0^z \gamma_{23} dz - \int_0^z \int_0^z \kappa_1 dz dz,
\tag{2b}
$$

$$
\delta_3(z) = \int_0^z \epsilon_{33} dz,
\tag{2c}
$$

$$
\theta_1(z) = \int_0^z \kappa_1 dz,
\tag{2d}
$$

$$
\theta_2(z) = \int_0^z \kappa_2 dz,
\tag{2e}
$$

$$
\theta_3(z) = \int_0^z \theta_3' dz,
\tag{2f}
$$

and the sectional loads—also known as internal loads—along the beam length can be shown, on the basis of statics, to be

$$
f_1 = F_1,
\tag{3a}
$$

$$
f_2 = F_2,
\tag{3b}
$$

$$
f_3 = F_3,
\tag{3c}
$$

$$
m_1 = M_1 - (L - z)F_2,
\tag{3d}
$$

$$
m_2 = M_2 + (L - z)F_1,
\tag{3e}
$$

$$
m_3 = M_3,
\tag{3f}
$$

where $F_1$, $F_2$, $F_3$, $M_1$, $M_2$, and $M_3$ are the tip shear forces, tip extension force, tip bending moments, and tip torsion moment, respectively.



In the simplest case, assuming constant properties and clamped boundary conditions at $z = 0$, the relationship between
deflections and loads at the tip can be derived through integration yielding a beam level compliance matrix, $[C]$, such that

$$
\begin{bmatrix} \delta_1 \\ \delta_2 \\ \delta_3 \\ \theta_1 \\ \theta_2 \\ \theta_3 \end{bmatrix} = \begin{bmatrix} C_{11} & C_{12} & C_{13} & C_{14} & C_{15} & C_{16} \\ C_{21} & C_{22} & C_{23} & C_{24} & C_{25} & C_{26} \\ C_{31} & C_{32} & C_{33} & C_{34} & C_{35} & C_{36} \\ C_{41} & C_{42} & C_{43} & C_{44} & C_{45} & C_{46} \\ C_{51} & C_{52} & C_{53} & C_{54} & C_{55} & C_{56} \\ C_{61} & C_{62} & C_{63} & C_{64} & C_{65} & C_{66} \end{bmatrix} \begin{bmatrix} F_1 \\ F_2 \\ F_3 \\ M_1 \\ M_2 \\ M_3 \end{bmatrix}
\tag{4}
$$

where the respective beam compliance coefficients can be linked to the cross-sectional compliance coefficients through a matrix
$[L]$, which contains polynomials of the length of the beam obtained through the integration step, such that

$$
\left[S\right]^v = \left[L\right]^{-1} \left[C\right]^v .
\tag{5}
$$

where $[S]^v$ and $[C]^v$ refer to the vectorised versions of the cross-sectional and beam level compliance matrices, respectively.
To determine the beam level compliance matrix six simulations are run, with each applying only one of the six individual tip
load cases. The resulting tip deflections can then be grouped and used to compute $[C]$ using

$$
\left[C\right] = \begin{bmatrix} \delta_{1,F_1} & \delta_{1,F_2} & \delta_{1,F_3} & \delta_{1,M_1} & \delta_{1,M_2} & \delta_{1,M_3} \\ \delta_{2,F_1} & \delta_{2,F_2} & \delta_{2,F_3} & \delta_{2,M_1} & \delta_{2,M_2} & \delta_{2,M_3} \\ \delta_{3,F_1} & \delta_{3,F_2} & \delta_{3,F_3} & \delta_{3,M_1} & \delta_{3,M_2} & \delta_{3,M_3} \\ \theta_{1,F_1} & \theta_{1,F_2} & \theta_{1,F_3} & \theta_{1,M_1} & \theta_{1,M_2} & \theta_{1,M_3} \\ \theta_{2,F_1} & \theta_{2,F_2} & \theta_{2,F_3} & \theta_{2,M_1} & \theta_{2,M_2} & \theta_{2,M_3} \\ \theta_{3,F_1} & \theta_{3,F_2} & \theta_{3,F_3} & \theta_{3,M_1} & \theta_{3,M_2} & \theta_{3,M_3} \end{bmatrix} \times \mathrm{diag} \begin{pmatrix} F_1 \\ F_2 \\ F_3 \\ M_1 \\ M_2 \\ M_3 \end{pmatrix}^{-1} .
\tag{6}
$$

If the simulations are run using tip loads of magnitudes of one, this relation simplifies down to

$$
\left[C\right] = \left[\delta\right] .
\tag{7}
$$

In combination with Equation (5) it is possible to calculate the cross-sectional compliance terms from the deflections either
along the beam or at the tip using

$$
\left[S\right] = \left[L\right]^{-1} \left[\delta\right] .
\tag{8}
$$

## 4 Results

### 4.1 Benchmarking

The comparison of the converged results from all tools for the baseline case, along with the run times for each analysis,
is presented in Table 7. All compliance values are normalised to the predictions of the quadratic solid element model in





**Table 7.** Prediction comparison for mid-section base case.

| Modelling Tool | Relative Compliance Term [-] | | | | Run-time [s] |
| | Shear $S_{22}$ | Bending $S_{44}$ | Coupling $S_{46}$ | Torsion $S_{66}$ | |
|---|---|---|---|---|---|
| BECAS | 1.208 | 1.002 | 1.000 | 0.998 | 19 |
| VABS | 1.208 | 1.002 | 1.000 | 0.998 | 8 |
| Lin. Shell | 1.728 | 0.990 | 0.992 | 1.314 | 99 |
| Quad. Shell | 1.439 | 0.990 | 0.985 | 1.195 | 222 |
| Lin. Solid | 0.901 | 0.994 | 0.991 | 0.987 | 377 |
| Quad. Solid | 1.000 | 1.000 | 1.000 | 1.000 | 487 |

ABAQUS, which, based of fundamental principles of solid mechanics, is deemed to be the most accurate representation of the cross-section. It is evident that the cross-sectional modellers are an order of magnitude faster, which was to be expected. However, it should be noted that the meshes were not optimised for convergence at minimum computational cost.

In terms of predicted compliance terms, both the linear and quadratic shell element models perform poorly across the board, consistent with the findings of previous studies Branner et al. (2007). On the other hand, there is excellent agreement between the cross-sectional modellers and the quadratic solid element models in ABAQUS, with good agreement also observed with the linear solid elements models. The only significant discrepancy is observed in the shear compliance term.

Further simulations found this discrepancy to be roughly consistent. As such, while the exact values of $S_{22}$ do not agree with that predicted by solid element models, the sensitivity trends of interest in this study can still be reliably determined. Furthermore, $S_{22}$ plays only a small role in actual bend-twist coupling behaviour and hence small errors in its sensitivities are considered acceptable in exchange for increased simplicity of speed of analysis. As BECAS and VABS agree with one-another, only a single set of results is presented even though both tools were used for all variations.

## 4.2 Web Placement

Due to the general multi-part approach to assembling wind turbine blades, the bonding of webs is a source of geometric variation. The placement of the web influences the development of the shear flow, especially in sections where there are multiple webs. As the shear flow in each cell of the cross-section is related to its enclosed area, a shift of the web's position can have a drastic effect on torsional compliance. Furthermore, shifting of the webs changes the width of the sections of unsupported skin, which influences the warping deformations under loading, again affecting the compliance of blade.

Web positions can be off-target in two distinct manners, the first being chordwise location and the second being their angle. To capture these two distinct possible variations, the contact points on the top and bottom surface for each of the webs are varied by up to $\pm 5\%$ of the chord length towards the leading edge or towards the trailing edge, where translation is achieved



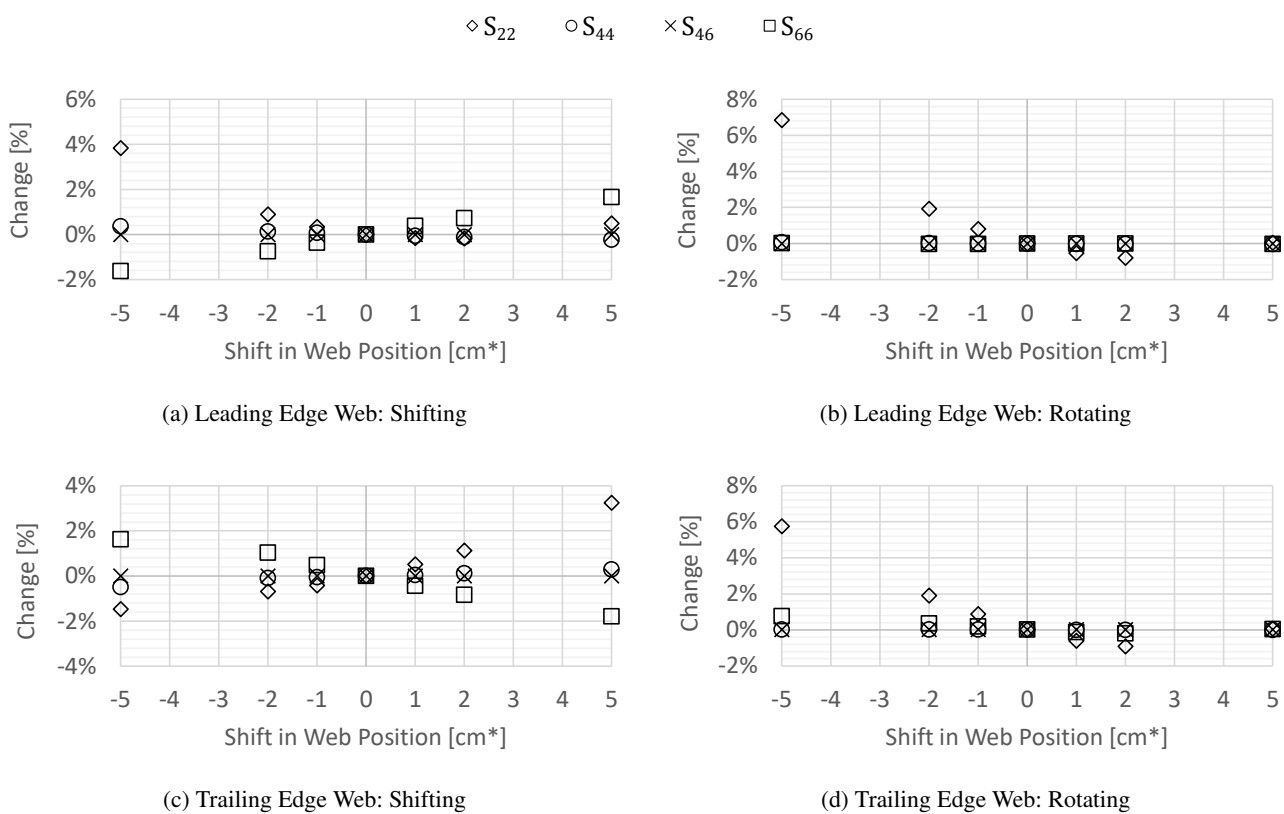

**Figure 3.** Variation in the compliance terms of thick-section due to changes in the web placement captured as motion of contact points for a chord length of $1\,\mathrm{m}$.

by moving both top and bottom surface contact points in the same direction and rotation is achieved by moving them in

215 opposing directions.

The results for all four webs—two in the thick-section and one each in the mid- and thin-section—are shown in Figure 3, Figure 4, and Figure 5. The main clear trend is that, apart from the shear term, $S_{22}$, the influence of web placement on compliance is actually minimal, with all other terms changing by no more than $2\,\%$ over the ranges investigated. The rotation of the webs, especially, has little to no effect in all terms aside from $S_{22}$. This can be explained by the fact that apart from

220 realigning the web to be more or less in line with the shear force, the rotation has little effect on the enclosed area of each cell, hence not meaningfully changing the shear flow within the cross-section.

The bending compliance, $S_{44}$ is largely unaffected by either shift or rotation, as the web contributes little to bending compliance. This can be traced to the BIAX layup in the webs, which gives it relatively little axial stiffness to combine with its small offset relative to the neutral axis. The fact that the bending compliance shifts slightly in the extremes of the range stems

225 from the fact that as the web is shifted and rotated it effectively changes size to match the inner contours of the skin, changing slightly the total area in the cross-section.



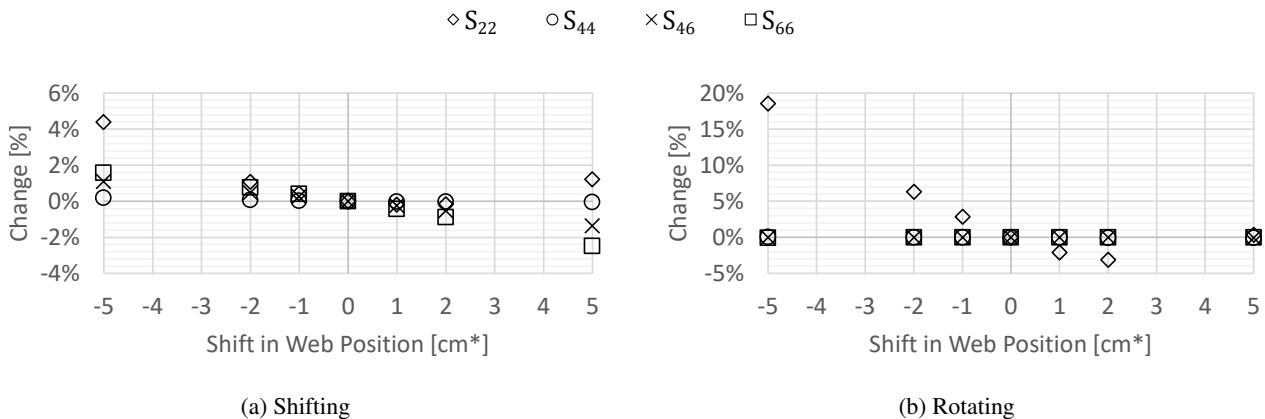

| (a) Shifting | (b) Rotating |

**Figure 4.** Variation in the compliance terms of mid-section due to changes in the web placement captured as motion of contact points for a chord length of 1 m.

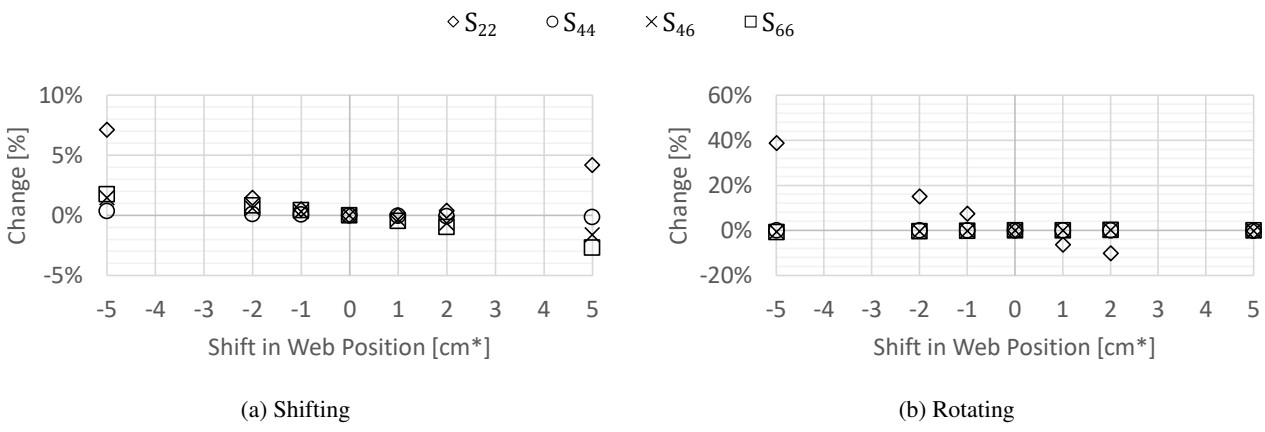

| (a) Shifting | (b) Rotating |

**Figure 5.** Variation in the compliance terms due to changes in the web placement captured as motion of contact points for a chord length of 1 m.





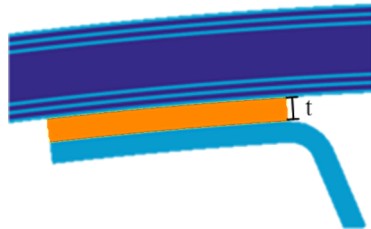

**Figure 6.** Close up of blade cross-section highlighting web-to-skin bondline thickness.

It is finally worth reflecting on the change to the coupling and torsional compliance caused by shifting of the web. This effect, which at the extremes of the changes can reach between $1\%$ and $2\%$, reflect the influence the web placements have on the distribution of shear flow between the different cells in the cross-section. From a design point of view, for the single web sections, industry trends pointed to pushing the web further back to reduce torsional and bend-twist coupled compliance. This can be traced both to the effect of reducing the size of the TE cell, as well as reducing the width of the TE panels, which reduces their ability to warp under loading and hence has a stiffening effect on the cross-section as a whole. Interestingly, for the thick-section with two webs, the data suggests that the leading edge web should be located further forward and the trailing edge web further backwards to reduce torsion compliance most significantly. This could, however, be a consequence of the starting positions of the webs, which were both ahead of the half chord point, which is common in wind turbine designs.

### 4.3 Web-to-Skin Bondline Thickness

The second manufacturing tolerance to be investigated is the bondline thickness between the web flange and both the upper and lower skins, see blade cross-section close-up in Figure 6. This is done with rather large margins using a variation of up to $50\%$ of the original bondline thickness. It should be noted that these changes are done without shifting the webs, hence this effectively also changes the height of the webs by the change in bondline thickness. Furthermore, as the contact points do not alter their positions, the change in bondline thickness also slightly changes the angle of the web. In spite of these variations clearly modifying the amounts of material present in the cross-section, the results as shown in Figure 7 demonstrate the limited impact this feature has on the predicted performance.

For all sections and all variations, the changes are within $1.5\%$. The largest sensitivity again occurs in the shear compliance, $S_{22}$. This can be traced to the minor rotation of the web induced by changing the bondline thickness without shifting the web. Previous results already demonstrated the sensitivity of the shear compliance to the angle of the web. The remaining terms are generally within $0.3\%$ across the whole range of different bondline thickness variations investigated. While critical for the strength performance of a blade, it is clear from these findings that stiffness performance is not sensitive to tolerances in bondline thicknesses between the webs and the skins.





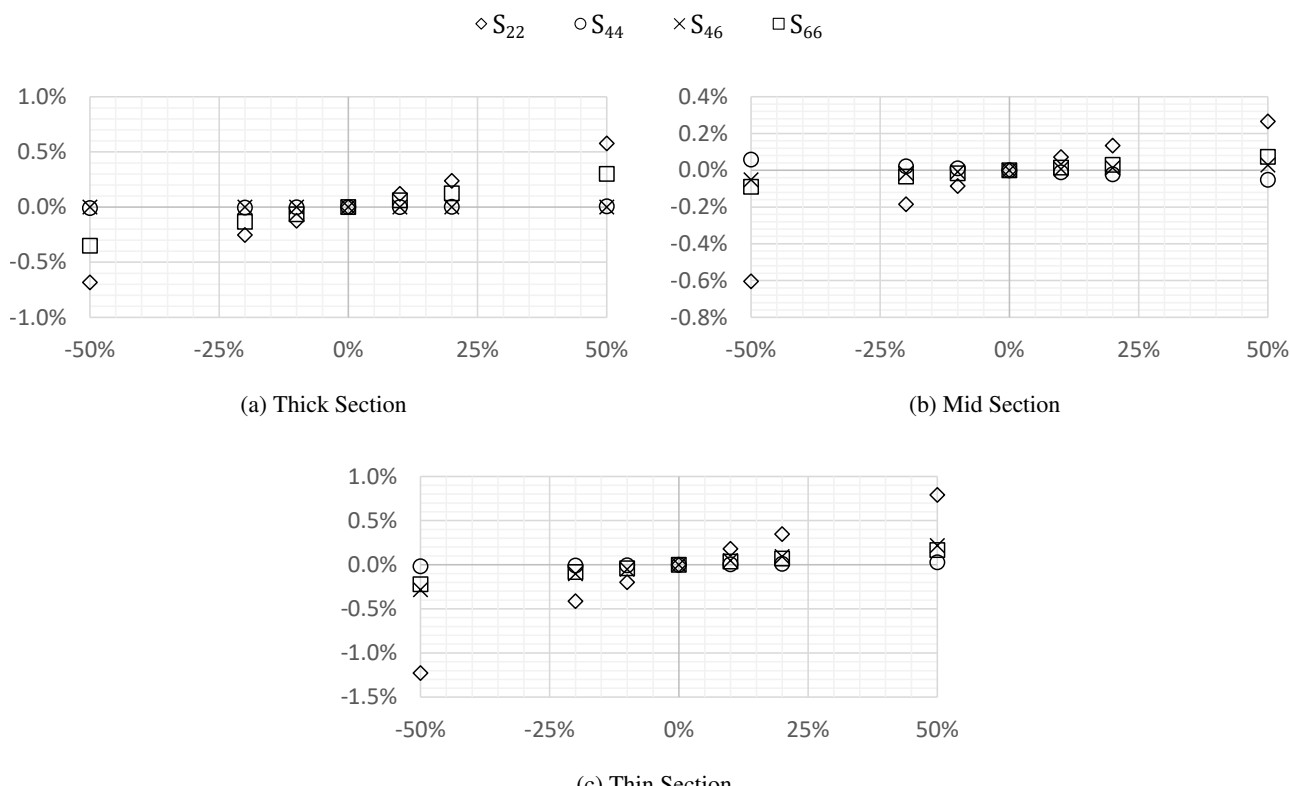

**Figure 7.** Variation in the compliance terms due to changes in the web bondline thickness.

### 4.4 Corner Radii

The next modelling variation to be explored is the corner of the webs. While manufacturing variations do exist in the corners of webs—especially considering the potential thermal spring back after cure—the biggest variation will come from instances where this feature is entirely ignored by the modelling approach. Here especially, it is worth noting that not only shell models tend to ignore this feature. Various example cross-sections in the Literature are depicted (and supposedly modelled) without a corner on the webs, see Figure 8 on page 512 in Chen et al. (2010) as an example. In these studies the cross-sections contain box beams or webs with no corners. Realistically speaking, however, it is known that these components are always produced with corners. As such, while a variation in corner radii is explored, see Figure 8, the main focus is on the comparison between a corner radii being present and no corner radii being present.

The results, shown in Figure 9, contain two distinct trends. The first is the overall lack of sensitivity, except for the thick-section, of the bending, coupling, and torsional compliance to all changes including the exclusion of the corner radius. The second is the pronounced—up to $9\,\%$ for the thin section—impact on the shear compliance when removing the corner radii.



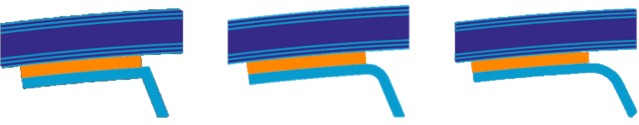

**Figure 8.** Cross-section detail showing range of corner radii from zero (left) through baseline (middle) to 50% larger (left).

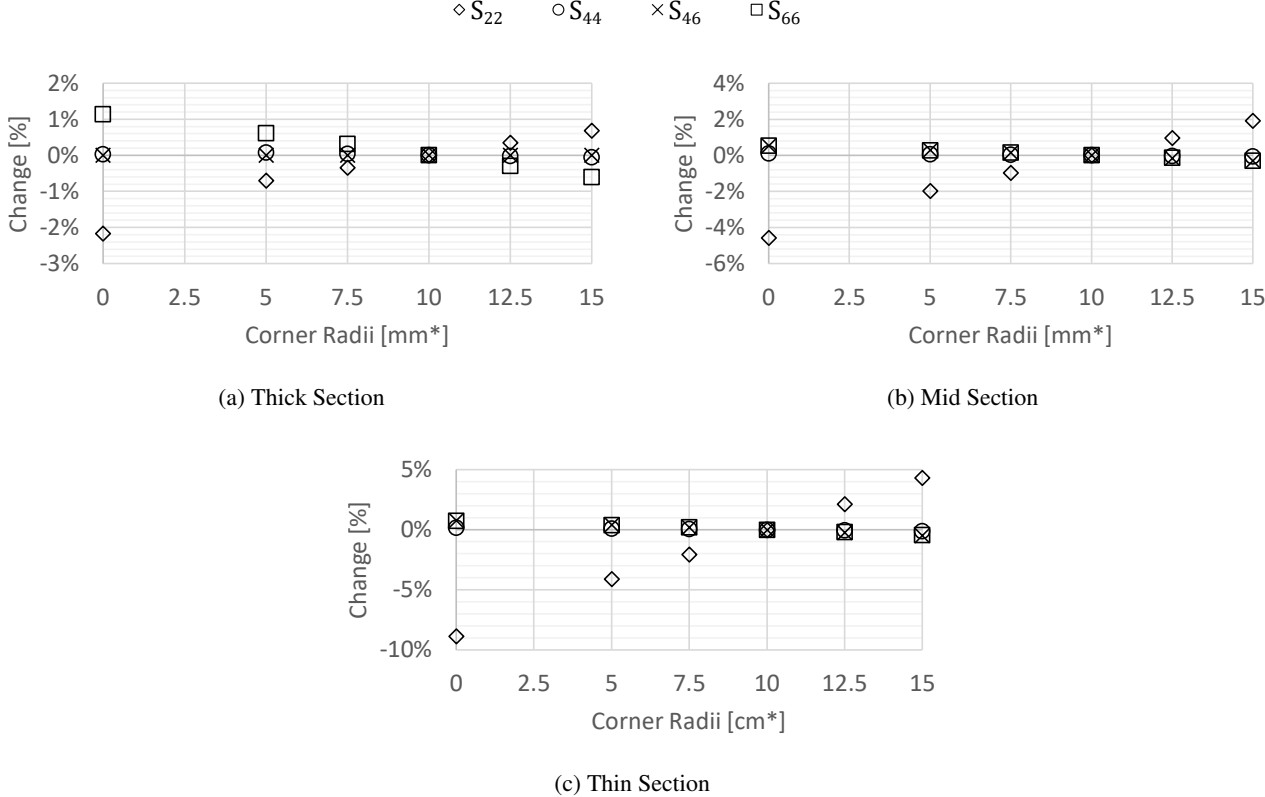

(a) Thick Section

(b) Mid Section

(c) Thin Section

**Figure 9.** Variation in the compliance terms due to changes in the web corner radius.

For the thin-section, where the webs constitute a larger portion of the total material in the cross-section, this change, combined with the slight rotation also caused, can explain the more pronounced influence on the shear compliance observed.

Finally, it is worth noting the relatively pronounced impact on the torsional stiffness in the thick-section. For this cross-
section, removing the corner radius increases the predicted torsional compliance by around $1\,\%$. While the changes for the three cross-sections analysed here are minor, the increased sensitivity in the torsional compliance case for the thick section poses the possibility that other design cases could show even larger sensitivities. The purpose of trialling three cases from different regions of the blade was to start to capture specific trends to highlight, in a broader design set, which variations may be critical to consider. The influence of the corner radii for the thick-section suggest potentially larger impacts for other design





cases. For instance, the sensitivity of the torsional term may grow with an increase in the number or relative placement of webs. Further trials on a wider set of baseline designs is needed to investigate this effect.

## 4.5    Trailing Edge Bondline Depth

The last modelling variation to be explored is the trailing edge bondline depth. Similar to the web corner radii, while there are genuine manufacturing tolerances at play, the largest predictive sensitivity stems from the way in which the feature is, or is

not, modelled. Realistically, wind turbine blades contain a trailing edge bondline, distinct from the rest of the material in the section. From a shear flow point of view, the depth of the TE bondline simultaneously changes the size of the TE cell as well as the width of the TE panel and the level of support/clamping it receives at the TE. As such, an increase in TE bondline depth can be expected to cause reduction in torsional compliance from both effects combining. It is also worth noting that the TE bond is a common point of failure in many blades. For these reasons, the trailing edge bondline is varied across a relatively

large scope, and also removed almost entirely, to assess the impact on the predicted properties.

From the plots in Figure 10, the influence of the TE bondline becomes readily apparent. Especially in the extreme case where it is reduced to a single element, the torsional compliance is $12\,\%$ higher for the thick-section, almost $18\,\%$ higher for the mid-section, and nearly $16\,\%$ higher for the thin-section. Even if the baseline depth is considered to be too large as a starting point, the difference from a base bondline of only $2.5\,\%$ of the chord length to a single element is still between $6\,\%$ and $8\,\%$.

These results were checked by recreating the single contact point scenario for the mid-section in 3D FEM using quadratic solid elements. The difference in the predicted torsional compliance between the cross-sectional modellers and the 3D FEM was $0.06\,\%$.

Beyond the effect on the torsional compliance, the results are in line with many of the other trends observed. The shear compliance shows a meaningful sensitivity to the changes, while the bending compliance shows almost no influence. Due to

the significantly larger variations in the torsion and shear terms, the variation in the coupling compliance of around $2\,\%$ from the baseline to the single contact point case is difficult to see. It is worth noting that the large variation in torsional compliance also means that the commonly used coupling stiffness coefficient Ong and Tsai (1999), which is a function of both direct and coupling compliance terms, reduces by around $7.5\,\%$ for the mid-section and $6\,\%$ for the thin-section.

The influence of the trailing edge bondline on compliance terms can be readily understood on the basis of the shear flow

in the trailing edge region. For the case of the baseline trailing edge depth against the minimum trailing edge bondline depth, when comparing the shear stress in the trailing edge region under unit bending loading, Figure 11 shows that there are two key differences. Firstly, in the baseline case, the shear stress in the upper surface, all of which is the consequence of BTC, drops to nearly zero shortly after the bondline begins. This is indicative of the shear stress transfer happening in the first few centimetres of the bondline. As such, the remaining area of the TE is effectively not carrying shear. It is also possible to note that the shear

stresses overall are much higher in the case of the minimum bondline depth. This reflects the reduction in support experienced by the upper and bottom surface, which leads to higher deformations and hence higher peak stress near the constrained point (*i.e.* the point where the surfaces are joined and hence restrict one another's motion). A final note can be made on the low shear stress in the bondline itself. This is due to its low stiffness resulting in stresses that are much smaller than those in the



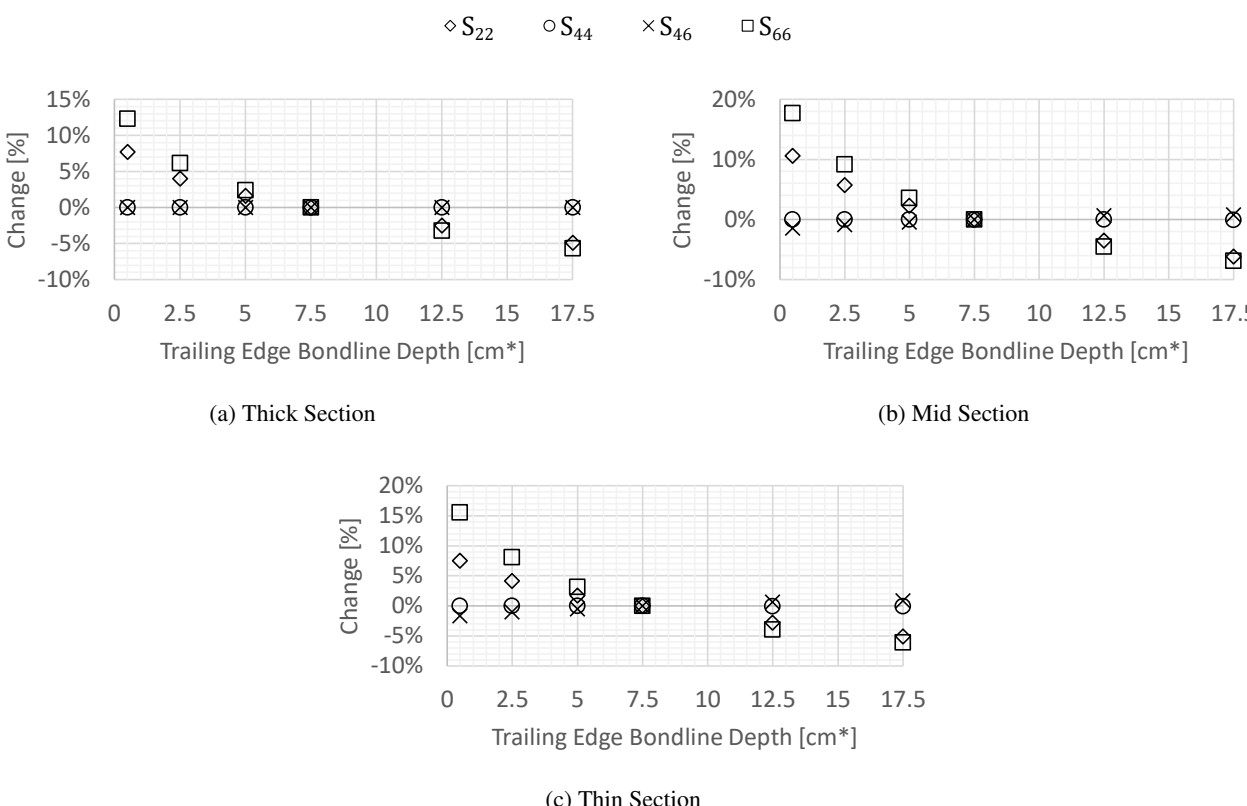

**Figure 10.** Variation in the compliance terms due to changes in the TE bondline depth.

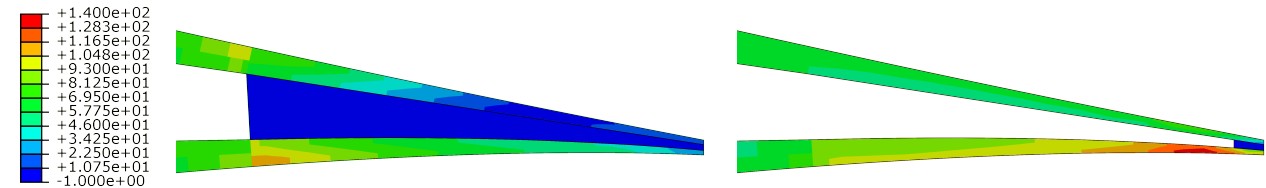

**Figure 11.** Shear stress plots, in $\mathrm{N\,m^{-2}}$, under bending loading, looking at the trailing edge region of the mid section for the base case (left) and the minimum bondline depth case (right). Results form ABAQUS quadratic solid element model.

composite laminate regions. The stresses in the bondline are only of the order of $0.1\,\mathrm{N\,m^{-2}}$, hence appearing to be zero using
the current legend.





## 5    Approximating Shell Element Predictions

Returning to the initial comparison between the modelling approaches, as documented in Table 7 for the mid-section base case, the shell models were shown to give fairly poor predictions. Based on the sensitivity data gathered, it appears that several features that are not properly resolved in the shell model could explain this error. To evaluate this, a comparison can be made

between the shell model and a solid model that approximates the 3D reality the shell model implies. In the shell model three key features are missing: the web corner radii, the web bondline, and—most importantly—the TE bondline.

When a solid model, using quadratic elements, was created with these features missing and/or misrepresented to agree with the shell model representation, the compliance terms showed similar deviations relative to the 3D FEM quadratic solid results for the actual base case. Specifically, the new normalised bending compliance was 1.003, the normalised coupling compliance

was 0.989, and the normalised torsional compliance was 1.170. This is close to the relative compliance value predicted by the original quadratic element shell model, which were 0.990 for the bending compliance, 0.985 for the coupling compliance, and 1.195 for the torsional compliance. The difference between the modified solid model and the shell model in terms of bending compliance is believed to be due to the thickness of the spar caps, a feature the shell model does not explicitly capture but the solid model does and which has already been shown to impact the shell model accuracy Branner et al. (2007). The new

comparison between shell and solid models made here adds to the existing Literature by highlighting a further limitations of shell models, where the accurate representation of the webs and the TE edge bondline need to be added to the list of features shell models do not adequately account for.

## 6    Discussion

In this work, the overall sensitivity of the shear, bending, coupling, and torsional compliance with regard to a selection of

manufacturing tolerances and modelling simplifications was captured for three representative cross-sections of a wind turbine blade. This was achieved by comparison of the predicted cross-sectional properties as provided by BECAS and VABS. To give credence to these results, for the mid-section, the base cases as well as one of the key variations were also carefully modelled in 3D FEM. For both these cases the 3D quadratic solid element models gave excellent agreement with the BECAS and VABS predictions. This was made possible in part by assuring the inputs provided to all tools were as near to identical as possible

within machine precision limits.

In its totality, the sensitivity study has provided two key findings regarding the compliance terms of coupled wind turbine blade sections. First, the compliance terms are mostly insensitive to the vast majority of manufacturing tolerances and modelling variations trialled. This finding should boost the confidence of industry in using BTC as it shows the technique can be used with the certainty that predicted performance can be achieved reliably. The key exception to this was the shear compliance,

$S_{22}$, which showed significant sensitivity to almost all variations. To the best of the authors' knowledge there are no studies that have isolated the influence of the shear compliance on the aeroelastic behaviour of wind turbine blades, so it is not known how this affects accurate aeroelastic performance predictions.





Aside from the shear compliance, some of the other terms also showed larger sensitivity in a select set of cases. In general, the compliance terms changed by much less than $1\%$ across what could be considered as the acceptable range of tolerances for the manufacturing variations investigated. In many instances, as the applied changes increased, the observed variations in the compliance terms grew rapidly, exceeding $5\%$ at extremes of the trialled domains. This is important to note when setting guidelines for manufacturing tolerances. However, it should be taken into account that all applied variations to dimensions were expressed as relative to the chord length. This means that at the extremes of most domains, the manufacturing tolerances being evaluated—given typical chord lengths on blades between $1\,\mathrm{m}$ and $5\,\mathrm{m}$—correlated to shifts of between $5\,\mathrm{cm}$ and $25\,\mathrm{cm}$. While, to the authors' best knowledge, no data on the manufacturing tolerances investigated in this study are available for real blades within the published literature, they are expected to be well within those limits numerically assessed here.

The second major finding of this numerical study relates to critical guidelines for modelling blade cross-sections and the overall fidelity and limitations of 3D shell modelling approaches for this purpose. While Branner *et al.* have explored this to some extent Branner et al. (2007), the only feature deemed important for the overall performances was the spar caps. The results presented in this article strongly suggest that, for accurately capturing the torsional performance, the model must fully resolve all features involved in the development of the shear flow within a cross-section. In this, 3D shell models suffer from an inability to properly capture the influence of the trailing edge bondline. Specifically, it was found that across all cases, when the trailing edge bondline depth was reduced, the torsional compliance increased significantly. This result was confirmed for the mid-section using a 3D quadratic solid element model which gave almost the exact same prediction of torsional compliance (difference of only $0.06\%$) compared to the cross-sectional modellers. A 3D quadratic solid element model was also used to further demonstrate that the inability of shell model to properly represent the trailing edge bondline, was responsible for the error in the torsion and coupling compliance terms for the shell models for the base case.

## 7 Conclusions

In summary, this study reviewed the influence of various geometric features on the accuracy of common cross-sectional modelling methods. Particular focus was placed on the features whose representation, based on common cross-sectional parameterisation approaches, can be ambiguous or even disregarded when using reduced geometrical models (*i.e.* shell elements). The findings highlight the importance of exercising careful control over inputs to ensure reliable predictions across different modelling techniques. Based on the results presented, the authors propose the following modelling guidelines:

1. Pure shell models, even for relatively thin sections, cannot be solely relied upon to accurately predict torsional compliance when significant bondlines are present.

2. Hybrid shell/solid models should incorporate solid elements not only in the spar cap areas, as previously demonstrated Branner et al. (2007), but also in the trailing edge region to capture the bondline and accurately predict the torsional behaviour of the structure.



3. Detailed features such as corner radii on webs and any fillets that similarly affect material distribution should be adequately resolved. Some allowances here can be made depending on acceptable level of fidelity.

4. All known manufacturing tolerances should be investigated to assess whether the design point exhibits sufficient robustness in its performance.

Understanding the limitations of numerical methods and establishing best practices for modelling is a crucial step in building confidence in the use of BTC. Future work should pursue experimental validation of the models, to demonstrate the accuracy of the models and the feasibility of implementing BTC in real-world applications.

*Data availability.* Data are available at the University of Bristol data repository, data.bris, at https://doi.org/10.5523/bris.o8a8gs1ee73y2ivi94338ghg4.

*Author contributions.* VKM: Conceptualization, Data Curation, Investigation, Methodology, Project administration, Validation, Visualization, Writing; TM: Conceptualization, Supervision, Writing - review & editing; PMW: Conceptualization, Funding acquisition, Supervision, Writing - review & editing; AP: Conceptualization, Funding acquisition, Supervision, Writing - review & editing.

*Competing interests.* This work was funded in part by Vestas Wind Systems A/S.

*Acknowledgements.* This research was supported by Vestas Wind Systems A/S and co-funded by Vestas Wind Systems A/S and the UK Engineering and Physical Sciences Research Council (EPSRC) through the Centre for Doctoral Training in Advanced Composites for Innovation and Science [Grant No. EP/L016028/1]. The support and funding is gratefully acknowledged.





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
