# Peer review of "Sensitivity of cross-sectional compliance to manufacturing tolerances for wind turbine blades"

_Wind Energy Science, 2023_

## Author Response (AR1)

**Referee Responses**

**Referee 1**

**General comments:**

This paper presents the calculation of cross-sectional blade properties, when certain geometrical dimensions are adapted, assumed to be representative for manufacturing tolerances.

Thank you for taking the time to review our paper and provide valuable feedback.

The presented cross-sectional shape is very simple and lacks realistic details, like ply drop-off's, fillets of adhesive joints, more complex lay-ups. The conclusion that a strongly reduced width of the bondline has the largest effect, is quite obvious and could be understood beforehand.

The referee is correct that, on first principles, the trends and correlations presented are obvious and conceptually well understood. Our intended take home message is not that we have discovered novel behaviour, but rather the extent and significance of it. Industry still uses simplified models and, in early design stages, struggles to capture cross-sectional torsional properties well. This means accurate early-stage aeroelastic designs are not possible. Our paper aims to provide designers with guidelines on cross-sectional modelling at the early design stages. For this we used an older blade platform as inspiration, which had a simpler design to start with. While a more complex design would introduce further manufacturing tolerances to explore and would result in different baseline values, the level of complexity in our models is sufficient for our aims. It's a level of simplification that does not compromise the analyses and conclusions presented.

The final lines of the abstract and final paragraph of the introduction have been modified to make our intentions for the paper more explicit and the final bullet point in the conclusion has been amended to acknowledge the realistic details raised by the referee as additional sources of manufacturing tolerances that should be accounted for.

**Detailed comments:**

1) several references are written duplicated in the text.

Thank you for pointing this out, this stemmed from the incorrect use of the LaTeX template on our end. All instances of this have been corrected.

2) the elastic constants in Table 3 contain errors. E12 cannot be equal to G13, and nu12, nu13 and nu23 cannot all have the same value. The Poisson's ratio for the adhesive seems also too low.

The typo of E12 has been corrected to G12.

Regarding the material properties, these values are approximations from actual material properties provided to us by Vestas, the true name and properties of the material cannot be disclosed due to an NDA. The values used are still in line with the actual material values. For the biaxial material specifically, the provided Young's modulus and shear stiffness in all direction were around 10GPa. Regarding the Poisson's ratios being the same in all directions, for isotropic materials (i.e. Foam and Adhesive) they would be identical in all directions. For the glass fibre composite layers they would indeed be different, however BECAS and VABS analysis are 2D cross-sectional simulations where the 23 terms are not active and the thin-walled nature of the cross-section means that through thickness terms have negligible impact even in the 3D analysis using ABAQUS.

We have split the lines in the table to avoid the implication that the noted terms are always the same and clarified the text explaining the origin of the property values.

3) it is not mentioned if the solid elements used are multi-layer elements, and no mesh convergence study is presented.

Additional details of the elements and the mesh resolution have been added to paragraph 2 of section 3. A new figure 3 has been added to demonstrate the convergence behaviour using the normalised bend-twist compliance term.

4) the authors seem to advocate Bend-Twist-Coupling, but do not mention about the thermal residual stresses and distortions that such unbalanced lay-ups might produce. That is also one of the reasons why BTC is not implemented in wind turbine blade industry.

The thermal residual stresses and distortions are indeed a consideration and paragraph 4 of Section 1 has been adjusted to acknowledge this. It is worth noting that the laminates used here are still symmetrical and so there are no B terms at the laminate level. Additionally, wind turbine blades are generally cured at low temperatures reducing the thermal stresses and distortions during manufacturing. Moreover, material coupling is not the only way to induce bend-twist coupling. Having consulted with numerous industrial experts, the uncertainty on the quantification of cross-sectional torsional stiffness has been identified as the main road blocker to the use of bend-twist coupling. This is reflected in the aims of our investigation.

5) nothing is mentioned about the exact modelling differences of the bondline between the different models. Mesh refinement cannot be seen from Figure 11, and the assumptions in the shell model are not listed in detail.

In the original solid models, the trailing edge bondline is modelled as a solid volume of the size shown in the left hand image in Figure 11. In the shell models, as the outer surface is used, the top and bottom skin only contact at the final node. In the modified solid model this is approximated reducing the bond line to the only the final element as shown in the right hand image in Figure 12. Additional comments have been added to section 5 to clarify this. We hope this addresses the referee's question.

6) the authors refer to old papers of Branner (2007) to justify their claim on too simplified modelling assumptions in the shell models, but those models are outdated and much better modelling approximations can be done now.

This is a fair point. Simulation capabilities have progressed significantly with the expansion of multi-scale methods as well as models such as the Carrera Unified Formulation. Industrial practices have, however, not caught up and many studies, especially early stage sizing, are still carried out using these more basic, "simplified" models. Our study looked to evaluate both the sensitivity of BTC terms to common manufacturing tolerances and in doing so provide insights to help bring industrial practice. As such, we believe the significant influence of trailing bondline thickness depth variation on cross-sectional properties demonstrated and the explanatory value this has for the limitations of an all-shell element model are worth publishing.

The relevant bullet point in the conclusions section has been amended to highlight the prevailing popularity of shell models in industry.

**Referee 2**

In this paper, the sensitivity of blade cross-sectional stiffness to the manufacturing tolerance (in terms of web placement, web-to-skin bondline thickness and corner radii) and the induced geometric variations was studied using different numerical tools. In general, the paper is well written.

Thank you for reviewing our paper and providing valuable feedback.

However, some details about the blade, the loading conditions and the analysis procedure could be provided. How long was the blade and what was the rated power for the turbine?

In this study we did not consider a full blade, but rather sample cross-sections. For the 3D models in

ABAQUS, nominal loadings (of value 1) were used to capture the deflections, which were then used to calculate the net cross-sectional properties using the equations derived in the paper. The cross-sections analysed in this study are not from a real blade, they are approximations derived from a blade under 100 meters in length and rated in the low MW range. More specific details cannot be shared for IP restrictions.

It seems that the complete blade was considered with tip loads. A drawing of the loaded blade and the results of the blade global deformation could be made. The authors should also explain what magnitude of the tip loads is considered and why. In principle, one could consider the maximum quasi-static loads when the turbine operates at the rated wind speed. The magnitude of the load will determine whether a linearized cross-sectional stiffness could be made or not. Is the tip load only representative for the actual loads? Would it be possible to apply the distributed lift and drag line forces along the blade, which are probably more realistic.

The results presented in this study are cross-sectional properties, which can be extracted from a 3D model using unit tip loads. However, no actual full blade simulations were run and so we cannot provide the information requested as it does not exist.

5) Would be the nonlinear stiffness behavior of the blade under the actual loads important?

In a full blade, the nonlinear stiffness behaviour would indeed be important under actual loading conditions. This is, however, beyond the scope of our current study.

---

## Author Response (AR2)

**Author Response**

**Referee Report #2**

I don't have more comments on the paper, except that the authors can give more details about the benchmarking of BECAS and VABS with the ABAQUS models, especially to show the loading conditions that are applied in ABAQUS.

Thank you for continuing to provide feedback on our manuscript. An additional schematic and paragraph have been added to Section 3 (page 10) to provide clarity on the analysis carried out in ABAQUS for benchmarking BECAS and VABS.